# DropCov: A Simple yet Effective Method for Improving Deep Architectures

**Qilong Wang**[1,3], **Mingze Gao**[1], **Zhaolin Zhang**[1], **Jiangtao Xie**[2], **Peihua Li**[2], **Qinghua Hu**[1,3,*]

[1]Tianjin University, China, [2]Dalian University of Technology, China,
[3] Haihe Laboratory of Information Technology Application Innovation, Tianjin, China
`qlwang@tju.edu.cn, gaomingze@tju.edu.cn, zzl9@tju.edu.cn`
`jiangtaoxie@mail.dlut.edu.cn, peihuali@dlut.edu.cn, huqinghua@tju.edu.cn`

## Abstract

Previous works show global covariance pooling (GCP) has great potential to improve deep architectures especially on visual recognition tasks, where post-normalization of GCP plays a very important role in final performance. Although several post-normalization strategies have been studied, these methods pay more close attention to effect of normalization on covariance representations rather than the whole GCP networks, and their effectiveness requires further understanding. Meanwhile, existing effective post-normalization strategies (e.g., matrix power normalization) usually suffer from high computational complexity (e.g., $O(d^3)$ for $d$-dimensional inputs). To handle above issues, this work first analyzes the effect of post-normalization from the perspective of training GCP networks. Particularly, we for the first time show that *effective post-normalization can make a good trade-off between representation decorrelation and information preservation for GCP, which are crucial to alleviate over-fitting and increase representation ability of deep GCP networks, respectively*. Based on this finding, we can improve existing post-normalization methods with some small modifications, providing further support to our observation. Furthermore, this finding encourages us to propose a novel pre-normalization method for GCP (namely DropCov), which develops an adaptive channel dropout on features right before GCP, aiming to reach trade-off between representation decorrelation and information preservation in a more efficient way. Our DropCov only has a linear complexity of $O(d)$, while being free for inference. Extensive experiments on various benchmarks (i.e., ImageNet-1K, ImageNet-C, ImageNet-A, Stylized-ImageNet, and iNat2017) show our DropCov is superior to the counterparts in terms of efficiency and effectiveness, and provides a simple yet effective method to improve performance of deep architectures involving both deep convolutional neural networks (CNNs) and vision transformers (ViTs).

## 1  Introduction

Global covariance pooling (GCP) has shown remarkable potential to improve performance of deep architectures in a variety of tasks, especially visual recognition [31, 23, 29, 30, 8, 48, 36, 47, 51]. One of core differences among various deep GCP methods is post-normalization for covariance representations, which plays a crucial role in final performance. The existing post-normalization approaches can be roughly categorized into element-wise post-normalization methods [31, 25, 2] and structure-wise ones [23, 29, 28, 25, 36, 47], which concentrate on normalizing entries and

---

*Qinghua Hu is the corresponding author and is with Engineering Research Center of City intelligence and Digital Governance, Ministry of Education of the People's Republic of China.

36th Conference on Neural Information Processing Systems (NeurIPS 2022).

eigenvalues of covariance matrices, respectively. Some researches (e.g., [30]) perform two kinds of normalizations simultaneously. Although several post-normalization strategies have been studied, behavior of different approaches seem vary significantly and there is a lack of an intuitive and unified interpretation. Therefore, the true mechanism lying behind these post-normalization methods (especially under deep architectures) is still an open problem.

Existing works understand the effect of post-normalization methods of GCP from different aspects. Specifically, [31, 25] claim that their element-wise normalization strategies can suppress the burstiness of covariance representations as in the classical bag-of-words [37] or actually perform a kind of whitening on representations. DeepO$_2$P [23] and MPN-COV [29] develop post-normalization methods to exploit geometry of covariances by considering the fact that space of covariance forms a Riemannian manifold [1, 34]. Besides, MPN-COV shows that matrix square-root (sqrt) normalization accounts for a robust covariance estimation [45], while [30] claims spectral normalization compensates for burstiness of eigenvalues and shows effectiveness of matrix sqrt normalization via empirical comparisons. Although many efforts are made, the aforementioned researches try to explain why GCP works by studying how post-normalizations affect covariance representations, which lack a full view on the whole GCP networks. Therefore, this work analyzes effect of post-normalization approaches on optimizing deep GCP networks. Particularly, by taking matrix power normalization [29] as an example, we find that post-normalization tries to handle a paradox between representation decorrelation and information preservation for GCP, which are crucial to alleviate over-fitting and increase representation ability of deep GCP networks, respectively. Particularly, effective post-normalization approaches can achieve a good trade-off, and so lead to a better performance. Based on this finding, we introduce some modified versions of existing post-normalization methods (e.g., adaptive power normalization for MPN-COV, I-LogM for DeepO$_2$P, and linear transformation for B-CNN [31]), which achieve further improvement, providing support to our observation.

Another issue of existing post-normalization methods is high computational complexity. Particularly, effective structure-wise post-normalization strategies usually involve singular value decomposition (SVD) of covariance matrices or sequential matrix multiplications, resulting in the computational complexity of $O(d^3)$ for $d$-dimensional inputs. Inspired by finding above, this paper proposes an efficient pre-normalization method for GCP (namely DropCov) based on dropout [26], as it has the natural ability to perform feature decorrelation [4] and reduce over-fitting of training neural networks [26]. Specifically, our DropCov develops an adaptive channel dropout on features right before computation of covariance representations, where probability of dropout is adaptively decided by seeking a good trade-off between representation decorrelation and information preservation. Compared to existing post-normalization methods, our DropCov achieves superior or competitive performance with a linear computational complexity of $O(d)$, while it is only used for training.

To verify the effectiveness of our proposed DropCov, we conduct experiments using various deep architectures including CNNs [17] and vision transformers (ViT) [43, 32, 50] on ImageNet-1K [7], ImageNet-C [18], ImageNet-A [19], Stylized-ImageNet [13], and long-tailed iNat2017 [20]. The contributions of this work are summarized as follows: (1) To our best knowledge, this work for the first time analyzes the effect of post-normalization on GCP from the perspective of optimizing neural networks, and we show that effective post-normalization methods can make a good trade-off between representation decorrelation (*w.r.t* over-fitting reduction) and information preservation (*w.r.t* increase of representation ability) for GCP (*w.r.t* deep GCP networks); (2) Based on our finding, we make some small modifications to existing post-normalization methods (Sec. 2.2), e.g., adaptive power normalization and I-LogM & linear transformation. The comparisons in Sec. 5.2 show our modifications can bring further improvement, giving support on our observation; (3) Profiting from our finding, we propose a novel DropCov method by developing an adaptive channel dropout as pre-normalization of GCP, which has a linear complexity of $O(d)$ and is only used for training. Extensive experiments show our DropCov is superior or competitive to existing post-normalization approaches and provides a simple yet effective method to improve deep architectures.

## 2 Effect of Post-normalization on Deep GCP Networks

In this section, we first analyze the effect of post-normalization on optimizing deep GCP networks by taking matrix power normalization as an example. To further verify our finding, we spread it to existing post-normalization approaches, and make some modifications according to our observations.

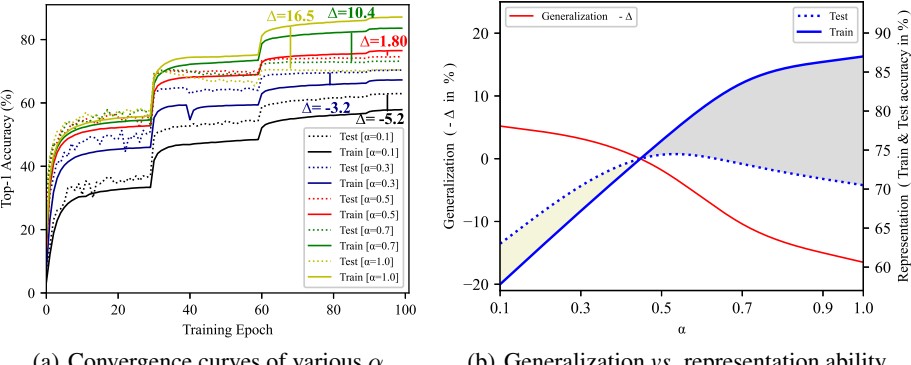

(a) Convergence curves of various $\alpha$      (b) Generalization *vs.* representation ability

Figure 1: Results of GCP networks with matrix power normalization (MPN) using the backbone of ResNet-18 ($d = 128$) on ImageNet-1K. (a) Convergence curves of various $\alpha \in (0, 1]$; (b) Comparison of generalization and representation abilities for various $\alpha$. From them we can see that larger values of $\alpha$ result in higher training accuracies indicated by blue solid line in (b), *w.r.t* better representation or approximation ability, but suffer from heavier over-fitting (*w.r.t* worse generalization), aka larger differences of training accuracy over test one indicated by $\Delta$. In contrast, smaller values of $\alpha$ have better generalization (aka smaller $\Delta$), while obtaining lower classification accuracies (*w.r.t* weaker representation ability). Particularly, $\alpha = 0.5$ achieves the best trade-off and so leads the best results.

## 2.1 How Does Matrix Power Normalization Impact GCP

**Matrix Power Normalization** Both previous works [47, 36] and our experiments show structure-wise post-normalization methods are generally superior to element-wise ones, where matrix power normalization (MPN) usually performs the most competitively. Let $\mathbf{X} \in \mathbb{R}^{N \times d}$ be $N$ $d$-dimensional features with zero-mean those outputted by the last convolution layer of deep CNNs or the last transformer block of ViT (i.e., final word tokens), GCP computes $\mathbf{X}^T\mathbf{X}$ with post-normalization to generate a covariance representation $\mathbf{Z}$. Specifically, GCP with MPN is calculated as

$$\mathbf{Z} = (\mathbf{X}^T\mathbf{X})^\alpha = \mathbf{U}\mathbf{\Lambda}^\alpha\mathbf{U}^T, \tag{1}$$

where $\mathbf{Z} \in \mathbb{R}^{d \times d}$ is used for final classification. $\alpha > 0$ is a parameter of power. $\mathbf{U}$ and $\mathbf{\Lambda}$ are the matrix of eigenvectors and the diagonal matrix of eigenvalues of $\mathbf{X}^T\mathbf{X}$, respectively.

**Effect of Power** For MPN, power $\alpha$ has a significant effect on the final results. Previous works [29, 30] conduct empirical comparison of $\alpha$, and choose the best one (e.g., $\alpha = 0.5$). However, the effect brought by $\alpha$ is still not very clear. In this work, we first analyze the effect of $\alpha \in (0, 1]$ on optimizing deep GCP networks. It is clear from Eqn. (1) that $\alpha$ has the following effect on GCP from the mathematical perspective:

$$\begin{cases} \mathbf{Z} \longmapsto \mathbf{I}, & \text{if} \quad \alpha \longmapsto 0 \quad : \text{Representation Decorrelation \& Information Loss,} \\ \mathbf{Z} \longmapsto \mathbf{X}^T\mathbf{X}, & \text{if} \quad \alpha \longmapsto 1 \quad : \text{Information Preservation \& Strong Correlation.} \end{cases} \tag{2}$$

Specifically, when $\alpha \longmapsto 0$, the normalized representation $\mathbf{Z}$ will be decorrelated and tend to be an identity matrix $\mathbf{I}$. Such representation decorrelation can help optimization of neural networks [27, 22, 4] (in particular for combating over-fitting [5]), which, however, will lead to information loss for $\mathbf{X}^T\mathbf{X}$, hurting representation ability of covariances. Particularly, $\mathbf{Z}$ with MPN of $\alpha = 0$ becomes an identity matrix, and the information lying in $\mathbf{X}^T\mathbf{X}$ completely loses. When $\alpha \longmapsto 1$, information of $\mathbf{X}^T\mathbf{X}$ will be gradually preserved, maintaining the correlations as characterized in the covariance matrices, which makes training of neural networks difficult (e.g., over-fitting). To further verify above observation, we conduct experiments using ResNet-18 on ImageNet-1K (more results can refer to supplementary materials). By observing Figs. 1(a) and 1(b), we draw the following conclusion:

- When $\alpha < 0.5$, post-normalization (i.e., MPN) tends to decorrelate representation. According to red line of Fig. 1(b), smaller values of $\alpha$ reduce over-fitting more significantly and have better generalization (aka smaller differences of training accuracy over test one [3, 5]). Meantime, indicated by blue lines of Fig. 1(b), smaller $\alpha$ leads lower accuracies (*w.r.t* weaker representation or approximation ability [35, 47]) due to information loss.

- When $\alpha > 0.5$, post-normalization (i.e., MPN) tends to preserve information. The training accuracies (*w.r.t* representation or approximation ability) of deep GCP networks indicated by blue solid line of Fig. 1(b) increase along values of $\alpha$ enlarge, but heavier over-fitting (aka larger values of $\Delta$) occur because effect of representation decorrelation decreases.

According to Eqn. (2) and above observation, we have

**Corollary 1.** *Effective post-normalization (e.g., matrix power normalization (MPN) with $0 < \alpha \leq 1$) can achieve a good trade-off between representation decorrelation and information preservation for GCP, which are crucial to alleviate over-fitting and increase representation ability of deep GCP networks, respectively. Particularly, MPN with $\alpha = 0.5$ achieves the best trade-off for $\alpha \in (0, 1]$ (without considering other factors), which is proved to be the widely used choice of $\alpha$ [29, 30].*

The Corollary 1 illuminates the effect of MPN on deep GCP networks from the perspective of model training, and gives an intuitive explanation about the choice of parameter $\alpha$.

## 2.2 Extension of Corollary 1 to Existing Post-normalization Methods

To further verify Corollary 1, we extend it to analyze behavior of existing post-normalization. Accompanied by these analyses, we can improve previous methods with some small modifications.

**Adaptive Power Normalization (APN)** Above all, Corollary 1 shows MPN with $\alpha = 0.5$ is the best trade-off for $\alpha \in (0, 1]$ if no other factor is considered. To further verify the effect of trade-off, we propose an adaptive power normalization (APN) by considering the effect of inputs (i.e., eigenvalues $\mathbf{\Lambda}$ of covariances). Specifically, we formulate the trade-off between representation decorrelation and information preservation as the following objective function:

$$\min_{\alpha} \log \left( \frac{\lambda_{max}^{\alpha}}{\lambda_{min}^{\alpha}} \right), \quad s.t. \ \max_{\alpha} \sum_i \frac{\lambda_i^{\alpha}}{\sum_i \lambda_i^{\alpha}} \log \left( \frac{\lambda_i^{\alpha}}{\sum_i \lambda_i^{\alpha}} \right), \tag{3}$$

where $\lambda_{max}$ and $\lambda_{min}$ are maximum and minimum of $\mathbf{\Lambda}$, respectively. Particularly, we minimize the ratio of $\lambda_{max}^{\alpha}$ to $\lambda_{min}^{\alpha}$ (in logarithmic coordinates) to enforce representation decorrelation. Here, smaller values of the objective function indicate stronger decorrelation, and $\alpha = 0$ achieves minimum (i.e., zero). On the other hand, we maximize negative entropy of normalized eigenvalues to keep discrepancy among eigenvalues and so preserve information, which achieves maximum value (i.e., zero) for $\alpha \to +\infty$. By considering that previous works [29, 30] show relation between performance and $\alpha$ is generally convex, we exploit a grid search strategy to obtain a suboptimal solution for $\alpha \in (0, 1]$ with the interval of $0.01^2$. However, $\lambda_{min}$ is usually nearby zero, leading to computational instability. Therefore, we optimize the objective function (3) as

$$\min_{\alpha} \Big[ \underbrace{\alpha(\log(\lambda_{max}) - \log(\max(C, \lambda_{min})))}_{\text{representation decorrelation}} \underbrace{- \tau \sum_{i=1}^{K} (\lambda_i^{\alpha} / \sum_{i=1}^{K} \lambda_i^{\alpha}) \log(\lambda_i^{\alpha} / \sum_{i=1}^{K} \lambda_i^{\alpha})}_{\text{information preservation}} \Big], \tag{4}$$

where we set minimum of $\lambda_{min}$ to a constant $C$ (e.g., 1e-2), and compute the entropy using Top-K eigenvalues. $\tau$ is a balance parameter, which is set to 1.2. Note that Eqn. (4) is computed efficiently for small $K$ ($K \leq 3$ in our work) using about a hundred trials, given the eigenvalues $\mathbf{\Lambda}$.

**MPN with $\alpha > 1$** Theoretically, $\alpha$ of MPN could be larger than one, which, however, violates the principle of representation decorrelation. Meanwhile, it disrupts equilibrium between eigenvalues, hurting information inherent in covariances. As illustrated in Fig. 2, MPN with $\alpha = 2$ suffers from heavy over-fitting, and is clearly worse than one with $\alpha = 0.5$.

**Matrix Logarithm Normalization (LogM)** LogM [23] is one another popular structure-wise post-normalization. Mathematically, logarithm has similar behavior with power function of $\alpha = 0.3$ when eigenvalues $\lambda > 1$, but reverses amplitudes of eigenvalues $\lambda < 1$ (i.e., neither decorrelation nor information preservation). However, small eigenvalues frequently occur in deep models [29], especially for large feature dimension ($d$). Accordingly, as shown in Fig. 2, LogM achieves similar

---

²Note that it is unnecessary to obtain exactly the optimal solution, since performance gap is negligible when difference of $\alpha$ is less than 0.01.

results with MPN of $\alpha = 0.3$ for $d = 128$. For $d = 256$, GCP network with LogM fails to converge due to side effect brought by small eigenvalues ($\lambda < 1$). *Therefore, we modified LogM with a shift operation (namely I-LogM) to make all inputs of LogM be larger than one, i.e.,* $\log(\lambda_i) \longrightarrow \epsilon \log((\lambda_i + \epsilon)/\epsilon)$. *Our I-LogM brings promising gains when $d = 256$* (see Sec. 5.2 and Table 6).

**Element-wise Normalization (EwN)** EwN could be roughly written in a unified formulation: $\boldsymbol{\beta} \frac{f(\mathbf{z}) - \mu}{\|f(\mathbf{z}) - \mu\|_*} + \boldsymbol{\gamma}$, where $\mathbf{z}$ is vectorization of $\mathbf{X}^T\mathbf{X}$. $f$ is a mapping function and $\|\mathbf{a}\|_*$ indicates a certain norm of $\mathbf{a}$. $\boldsymbol{\beta}$ and $\boldsymbol{\gamma}$ perform a linear transformation (LT). Particularly, for element-wise power normalization (EPN) [31], $f$ is a signed power function and $\mu = 0$. $\|\mathbf{a}\|_*$ is a $\ell_2$ norm, and no LT is performed. For LN [2], $f$ is an identity mapping and $\mu$ is mean of $f(\mathbf{z})$. $\|\mathbf{a}\|_*$ is a $\ell_2$ norm. According to above formulation and Corollary 1, EwN performs representation decorrelation based on zero-mean (i.e., $f(\mathbf{z}) - \mu$) and standardization (i.e., normalized by $\|f(\mathbf{z}) - \mu\|_*$), while LT

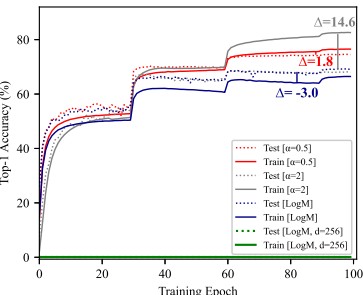

Figure 2: Results of $\alpha > 1$ & LogM

operation aims to recover information. Compared to MPN, element-wise normalization lacks an effective mechanism to balance representation decorrelation and information preservation. Particularly, as shown in Sec. 5.2, EPN does not work well, since only representation decorrelation is performed. *Therefore, we introduce a LT for EPN, and it achieves promising improvement* (see Table 6).

## 3 DropCov: A Simple yet Effective Normalization

### 3.1 Normalizing GCP via Dropout

Corollary 1 shows the core effect of effective post-normalization (e.g., MPN) lies in a good trade-off between representation decorrelation and information preservation for optimizing deep GCP networks. However, MPN suffers from high computational complexity (i.e., $O(d^3)$ of $d$-dimensional features), especially for high-dimensional inputs. Previous works [41, 5, 4] show that dropout can reduce correlation among features. Therefore, one question is naturally raised up: *Could we use efficient dropout to perform representation decorrelation for GCP?* Given probability $\rho$ of dropout, $\rho \to 1$ leading to sparser, less correlated features but more information loss. On the contrary, $\rho \to 0$ preserves more information while performing less decorrelation. Above observations show consistence with Corollary 1. However, there exist two issues to apply dropout for normalizing GCP: (1) how to perform dropout and (2) how to choose probability $\rho$ of dropout. A naive method is to perform dropout with a fixed $\rho$ after covariance representations (as compared in Table 2), but it has several limitations: first, dropout after covariance representations will break their structures (e.g., row $i$ of covariance means correlations among $i$-th channel and all ones); second, fixed $\rho$ hardly achieves trade-off between representation decorrelation and information preservation; third, complexity of such strategy is quadratic to feature dimension $d$ (i.e., $O(d^2)$), and is still high for large $d$.

### 3.2 Pre-Normalization via Adaptive Channel Dropout (ACD)

To handle above issues, this work proposes a pre-normalization method (namely DropCov) by performing channel dropout with an adaptive probability $\rho$ on features $\mathbf{X}$ before GCP, which aims to reach a good trade-off between representation decorrelation and information preservation. Since DropCov performs representation decorrelation via an adaptive dropout operation on channel features before GCP, it can maintain structure of covariance representations, while having a more efficient, linear computational complexity of $O(d)$. To reach the trade-off with adaptive probability $\rho$, we compute it by considering two factors: (1) feature dimension ($d$) and (2) relationship between feature correlation and feature importance. Then, we have

$$\rho = 1 - \frac{D}{\log(d)}\left(\frac{\boldsymbol{\omega}^T\boldsymbol{\pi}}{\|\boldsymbol{\omega}\|\|\boldsymbol{\pi}\|}\right), \tag{5}$$
$$\boldsymbol{\omega} = \sigma(\text{C1D}_3(\text{GAP}(\mathbf{X}))), \quad \boldsymbol{\pi} = \text{SUM}_{\text{row}}(\mathbf{X}^T\mathbf{X}),$$

where $\boldsymbol{\omega}$ and $\boldsymbol{\pi}$ represent feature importance and feature correlation, respectively. $D$ is a scaling parameter, which is set to 10 throughout all experiments. Since self-attention mechanisms [21, 46] are

Table 1: Comparison of our DropCov with close related works in terms of formulation, computational complexity of post-normalization and usage mode. Please refer to Sec. 4 for the details of symbols.

| | Method | Formulation | Complexity | Train | Infer. |
|---|---|---|---|---|---|
| Element-wise Post-Norm. | B-CNN [31] | $\ell_2(sqrt(\text{V}(\mathbf{X}^T\mathbf{X})))$ | $O(d^2)$ | ✓ | ✓ |
| | SigmE [25] | $2/(1 + e^{-\beta \cdot \text{V}(\mathbf{X}^T\mathbf{X})}) - 1$ | $O(d^2)$ | ✓ | ✓ |
| | LN [2] | $\boldsymbol{\beta} \odot (\text{V}(\mathbf{X}^T\mathbf{X}) - \mu)/\sigma) \oplus \boldsymbol{\gamma}$ | $O(d^2)$ | ✓ | ✓ |
| Structure-wise Post-Norm. | DeepO$_2$P [23] | $\text{V}(\text{LogM}(\mathbf{X}^T\mathbf{X}))$ | $O(d^3)$ | ✓ | ✓ |
| | MPN-COV [29] | $\text{V}((\mathbf{X}^T\mathbf{X})^\alpha)$ | $O(d^3)$ | ✓ | ✓ |
| | IB-CNN [30] | $\ell_2(sqrt(\text{V}(\mathbf{X}^T\mathbf{X})^{1/2}))$ | $O(d^3)$ | ✓ | ✓ |
| | iSQRT-COV [28] | $\text{V}(\approx (\mathbf{X}^T\mathbf{X})^{1/2})$ | $O(d^3)$ | ✓ | ✓ |
| | MaxExp [25] | $\mathbf{I} - (\mathbf{I} - \mathbf{X}^T\mathbf{X}/(\mathbf{X}^T\mathbf{X} + \varepsilon))^\alpha$ | $O(d^3)$ | ✓ | ✓ |
| Ours | DropCov | $\text{V}(\mathbf{Y}^T\mathbf{Y}), \ \mathbf{Y} = \delta_\rho(\mathbf{X})$ | $O(d)$ | ✓ | × |

✻: Note $\beta$ and $\varepsilon$ are parameters. $\odot$ and $\oplus$ indicate element-wise multiplication and addition, respectively.

usually used to search important features, we measure feature importance by modifying a light-weight channel attention module [46], which involves global average pooling (GAP), a 1D convolution with kernel size of 3 (C1D$_3$), and softmax function ($\sigma$). As covariance $\mathbf{X}^T\mathbf{X}$ naturally characterizes correlation among different features, we compute feature correlation by summarizing the elements along row of $\mathbf{X}^T\mathbf{X}$ (i.e., $\text{SUM}_{\text{row}}(\mathbf{X}^T\mathbf{X})$). Note that we restrict $\rho$ of Eqn. (5) belongs to $(0,1)$.

As for Eqn. (5), we adaptively decide probability $\rho$ of channel dropout for reaching a good trade-off between representation decorrelation and information preservation, where $\frac{D}{\log(d)}$ and inner product of $\langle \boldsymbol{\omega}, \boldsymbol{\pi} \rangle$ are designed to consider effect of feature dimension ($d$) and relationship between feature correlation and feature importance, respectively. In particular, larger feature dimension $d$ usually requires larger dropout probability $\rho$. Meanwhile, feature correlation (i.e., $\boldsymbol{\pi}$) and feature importance (i.e., $\boldsymbol{\omega}$) are closely related to representation decorrelation and information preservation, respectively. Clearly, larger feature correlation results in stronger representation correlation, while features with larger channel weights contain more important information. Therefore, relationship between feature correlation and feature importance is good indicator to perform channel dropout. Intuitively, if feature correlation is closely related to (i.e., seriously decoupled with) feature importance, it is hard to select features for reaching a good trade-off between representation decorrelation and information preservation, and so we need to carefully adopt dropout for channel features under the random setting, leading to a small $\rho$. Otherwise, we can perform dropout more safely to achieve a trade-off, and adopt a large $\rho$. Note that we show behavior of $\rho$ during training in supplementary materials. Based on Eqn. (5), our DropCov is achieved by

$$\mathbf{z} = \text{V}(\mathbf{Y}^T\mathbf{Y}), \ \mathbf{Y} = \delta_\rho(\mathbf{X}), \tag{6}$$

where $\delta_\rho$ is a channel dropout operation with the probability of $\rho$. V is a vectorization operation followed by triangulation, leading to a $d(d+1)/2$-dimensional covariance representation $\mathbf{z}$. Note that our DropCov (6) is only adopted to train GCP networks, which can be easily implemented by standard back-propagation. During inference stage, we simply use $\text{V}(\mathbf{X}^T\mathbf{X})$ for final classification.

## 4 Related Works

In this section, we review some works closely related to our DropCov. Table 1 gives a detailed comparison of our DropCov with existing post-normalization methods in terms of formulation, computational complexity of post-normalization and usage mode. For element-wise post-normalization methods, B-CNN[3] [31] performs element-wise signed square-root ($sqrt$) followed by a $\ell_2$ normalization on covariance representations. [25] proposes to use some surrogate functions (e.g., SigmE) to handle issue of negative evidence inherent in power function. Layer normalization (LN) [2] is proposed to normalize each feature with its mean ($\mu$) and variance ($\gamma$) followed by a linear transformation. Since these

---

[3]Note there exist small differences between GCP and bilinear pooling of B-CNN. Specifically, GCP computes covariance of inputs $\mathbf{X}$ with mean of $\boldsymbol{\mu}$, while bilinear pooling computes outer product of inputs $\mathbf{X}$ and $\mathbf{Y}$. When $\mathbf{X}$ and $\mathbf{Y}$ are shared and zero-mean, bilinear pooling captures the same information with one of GCP.

methods perform normalization on each element of covariance representations, they have the computational complexity of $O(d^2)$ for $d$-dimensional features $\mathbf{X}$. For structure-wise post-normalization methods, DeepO$_2$P and MPN-COV [29] perform matrix logarithm (LogM) and matrix power function for covariance representations, respectively. IB-CNN [30] combines matrix square-root normalization with element-wise one, while iSQRT-COV [28] proposes an approximate matrix square-root normalization via fast iterative matrix multiplications. Song et al. further analyze the effect of iSQRT-COV [38] and develop a faster variant [39]. [25] introduces several spectral power functions (e.g., MaxExp) for normalizing covariance matrices. These structure-wise approaches involve SVD of covariances or sequential matrix multiplications, having the computational complexity of $O(d^3)$. ReDro [36] proposes a relation dropout scheme to divide a large-sized covariance into a group of small-sized ones, aiming to mitigate the computational issue of matrix normalization [29, 30, 28, 11]. ReDro reduces computational complexity of structure-wise approaches from $O(d^3)$ to $O(d^3/G^2)$, where $G$ is the number of groups. Besides, all existing works perform post-normalization methods during both training and inference (Infer.) stages. Different from aforementioned works, our DropCov performs adaptive channel dropout $\delta$ on features $\mathbf{X}$ right before GCP, which has a linear complexity $O(d)$ and is free during inference stage. Experimental results in Sec. 5.3 show our DropCov with much less computational complexity is superior or competitive to previous works.

# 5  Experiments

In this section, we conduct experiments to evaluate effectiveness of our DropCov in improving deep architectures on image classification tasks. Specifically, we first describe implementation details, and then make ablation study on ImageNet-1K (IN-1K) [7] using backbone of ResNet-18. Additionally, we apply our DropCov to both ResNets [17] and ViT models [43, 32, 50], while comparing them on several benchmarks (i.e., IN-1K, ImageNet-C (IN-C) [18], ImageNet-A (IN-A) [19] and Stylized-ImageNet (Sty.-IN) [13]) to assess generalization and robustness of our DropCov. Finally, we assess generalization of DropCov models by transferring them to long-tailed species classification [20].

Table 2: Comparisons (% in Top-1 accuracy) of adaptive normalization methods with the fixed ones using ResNet-18 on ImageNet-1K.

| Method | $d = 64$ | $d = 128$ | $d = 256$ |
|---|---|---|---|
| DropElement ($\rho = 0.5$) | 73.4 | 74.6 | 74.0 |
| DropChannel ($\rho = 0.5$) | 70.1 | 73.1 | 75.1 |
| ACD (Ours) | **73.5** | **75.0** | **75.2** |
| MPN ($\alpha = 0.5$) | 73.1 | 74.4 | 74.9 |
| APN (Ours) | 73.3 | 74.5 | 75.0 |

Table 3: Comparisons (% in Top-1 accuracy) of several specific dropout strategies using ResNet-18 on ImageNet-1K.

| Method | $d = 64$ | | $d = 256$ | |
|---|---|---|---|---|
| | Element | Channel | Element | Channel |
| Large ($\rho = 0.5$) | 72.2 | 71.3 | 71.4 | 70.9 |
| Small ($\rho = 0.5$) | 68.4 | 71.2 | 66.4 | 71.5 |
| Uniform ($\rho = 0.5$) | 72.8 | 72.1 | 74.0 | 74.4 |
| ACD (Ours) | **73.5** | | **75.2** | |

Table 4: Comparisons (% in Top-1 accuracy) of various dropout variants using ResNet-18 on ImageNet-1K.

| Method | dim=64 | dim=256 |
|---|---|---|
| Maxout [15] | 72.1 | 73.7 |
| Dropconnect [44] | 70.6 | 72.5 |
| Decov [5] | 72.7 | 74.0 |
| Maxdropout [9] | 72.0 | 70.1 |
| DropBlock [14] | 72.3 | 74.1 |
| ACD(Ours) | **73.5** | **75.2** |

Table 5: Comparisons (% in Top-1 accuracy) of DropElement and DropChannel with various drop rates ($\rho$) using ResNet-18 on ImageNet-1K.

| | Channel Dropout | | Element Dropout | |
|---|---|---|---|---|
| $\rho$ | dim = 64 | dim = 256 | dim = 64 | dim = 256 |
| 0.1 | 72.5 | 71.9 | 72.3 | 70.8 |
| 0.3 | 72.8 | 74.7 | 72.8 | 72.3 |
| 0.5 | 70.1 | 75.1 | 73.4 | 74.0 |
| 0.7 | 65.7 | 72.1 | 72.1 | 74.2 |
| 0.9 | 20.5 | 54.7 | 68.2 | 73.8 |
| ACD (Ours) | **73.5** | **75.2** | **73.5** | **75.2** |

## 5.1  Implementation Details

In this work, we apply our DropCov to several ResNets [17] and ViT models [43, 32, 50]. Specifically, we construct DropCov models based on ResNet following exactly the same schemes in [29, 47]. Given a CNN backbone, we first introduce a $1 \times 1$ convolution after the last convolution layer for reducing dimension of features to $d$, and then replace the original GAP by our DropCov followed by

a fully-connected (FC) layer and softmax for classification. All models are optimized using the same hyper-parameter settings and data augmentations as suggested in [17, 29], where stochastic gradient descent (SGD) with initial learning rate ($lr$) of 0.1 is used to train the networks within 100 epochs and $lr$ is decayed by 10 every 30 epochs. We combine DropCov with ViT models as in [49], where one FC layer is employed to reduce dimension of final word tokens, and our DropCov replaces GAP and classification token for Swin Transformers [32] and DeiT/T2T-ViT [43, 50], respectively. These models are trained by using the same hyper-parameter settings in [43, 32, 50]. All programs run a sever equipped with eight Nvidia RTX 3090 GPUs and 128G RAM. Source code (e.g., PyTorch [33], PaddlePaddle and Mindspore) will be available at `https://github.com/mingzeG/DropCov`.

## 5.2 Ablation Study

In this subsection, we conduct experiments on ImageNet-1K using lightweight ResNet-18 to assess effect of adaptive probability $\rho$ of dropout on our method, while compare with several dropout strategies, and finally assess effect of our modifications on existing post-normalization approaches.

**Fixed $\rho$ vs. ACD.** Our DropCov performs pre-normalization for GCP by using an adaptive channel dropout (ACD) method, which automatically determines probability $\rho$ of dropout by considering trade-off between representation decorrelation and information preservation. We compare with two kinds of baseline dropout methods using fixed $\rho = 0.5$, i.e., element-wise dropout for covariance representations (DropElement) and channel dropout for input features (DropChannel). As shown in the upper part of Table 2, our ACD achieves the best results for various feature dimensions ($d$) and clearly outperforms the naive DropElement in Sec. 3.1. Besides, we compare with DropElement and DropChannel with various dropout ratios. As compared in Table 5, our ACD is superior to DropElement and DropChannel for all dropout ratios. Particularly, the best dropout ratios are quite different for two methods and various feature dimensions. In contrast, our ACD can adaptively determine probability $\rho$ of dropout by balancing representation decorrelation and information preservation, while always achieving the best performance, clearly verifying ACD can achieve a better trade-off.

**Comparison of Various Dropout Strategies.** To verify the effectiveness of our ACD, we compare with several dropout strategies. First, we perform element-wise (Element) dropout on some specific entries of covariance with $\rho = 0.5$. Specifically, we rank all entries of covariance in a descending order, and then perform dropout on top/bottom/uniform half of the ordered entries (namely Large/Small/Uniform). We adopt similar strategies to feature channels (Channel), where channel indicators are decided by attention module of our ACD. Note that 'Element' dropout preforms dropout on elements of covariance representations based on values of elements ($[\mathbf{X}^T\mathbf{X}]_{ij}$), which indicates correlation between $i$-th channel and $j$-th channel. Therefore, it only considers feature correlation. For 'Channel' dropout, it preforms dropout on feature channels based on their weights, which are obtained by attention module of ACD (i.e., $\boldsymbol{\omega}$). Therefore, it only considers feature importance. Based on above discussion, 'Element' and 'Channel' dropout methods can be regarded as special cases of ACD, where only feature correlation or feature importance are considered, respectively. As compared in Table 3, uniform dropout clearly outperforms those only focus on largest and smallest entries, indicating equilibrium of feature correlation or importance is crucial to final performance. Furthermore, our ACD is superior to uniform dropout by a clear margin. Besides, we compare with several existing Dropout variants, including Maxout [15], DropConnect [44], Decov [5], maxdropout [9] and DropBlock [14]. As shown in Table 4, our ACD clearly outperforms them. Although these methods can prevent overfitting, they are not good at making a balance between representation decorrelation and information preservation. These results further demonstrate the effectiveness of our ACD.

**Effect of Modifications.** According to Corollary 1, we make several modifications to existing post-normalization approaches. Beyond MPN, we propose an APN method by considering eigenvalues $\boldsymbol{\Lambda}$ of inputs (see Eqns. (3) and(4)). The results are compared in the bottom part of Table 2, where our APN determines $\alpha$ by considering effect of inputs while average values of $\alpha$ are $0.53$, $0.49$ and $0.45$ for $d = 256$, $d = 128$ and $d = 64$, respectively. Clearly, average values of $\alpha$ achieved by APN are nearby 0.5, which further account for why 0.5 is the widely used choice of $\alpha$ for MPN. Besides, APN brings 0.1%~0.2% gains over MPN with $\alpha = 0.5$ by considering effect of inputs. We would like to clarify they are non-trivial gains on ImageNet strong MPN, and the recent works [38, 40] also bring similar gains (0.1%~0.3%) over MPN. Besides, as discussed in Sec.2.2, we introduce an I-LogM and a linear transform (LT) based on BN [22] for improving LogM normalization (DeepO$_2$P)

Table 6: Comparison of DropCov with several post-normalization in terms of Top-1 accuracy (Acc.) and running speed using ResNet-18 on ImageNet-1K. $\star$: According to Sec.2.2 and Corollary 1, we modify B-CNN and IB-CNN with a linear transform (LT), while modifying DeepO$_2$P with I-LogM.

| Method | $d = 64$ | | | $d = 256$ | | |
|---|---|---|---|---|---|---|
| | Top-1 Acc. (%) | Train (μs) | Test (μs) | Top-1 Acc. (%) | Train (μs) | Test (μs) |
| Plain GCP | 71.1 | 3.45 | 1.04 | 70.0 | 35.08 | 11.89 |
| B-CNN [31] | 38.3 | 4.31 | 1.23 | 41.1 | 44.97 | 14.38 |
| B-CNN + LT$^\star$ | 68.3 | 4.45 | 1.28 | 73.2 | 47.37 | 15.24 |
| LN [2] | 71.7 | 4.10 | 1.13 | 70.2 | 40.79 | 13.59 |
| DeepO$_2$P [23] | 70.1 | 922.50 | 910.95 | *Not Converge* | 9731.71 | 9638.71 |
| I-LogM$^\star$ | 71.2 | 922.53 | 910.98 | 72.0 | 9732.46 | 9639.65 |
| MPN-COV [29] | 73.1 | 925.16 | 913.87 | 74.9 | 9735.11 | 9642.43 |
| iSQRT-COV [28] | 73.4 | 12.35 | 4.94 | 75.2 | 193.65 | 77.46 |
| IB-CNN [30] | *Not Converge* | 13.93 | 5.16 | 36.1 | 202.38 | 80.55 |
| IB-CNN + LT$^\star$ | 70.0 | 14.16 | 5.21 | 72.8 | 207.48 | 82.98 |
| DropCov (Ours) | **73.5** | **3.54** | **1.04** | **75.2** | **36.20** | **11.89** |

◇: Note we compute running speed (μs) of single GCP module with post-normalization on a 2080Ti GPU.
♯: The original ResNet-18 with GAP achieves 70.2% in Top-1 accuracy.

and element-wise post-normalization (i.e., B-CNN and IB-CNN), respectively. As shown in Table 6, DeepO$_2$P performs poorly troubled by small eigenvalues (especially for $d = 256$), while our I-LogM obtains promising gains for both low- and high-dimensional features. Furthermore, B-CNN and IB-CNN achieve very unsatisfied results; in contrast, our simple modification brings significant improvement for B-CNN and IB-CNN. Above results clearly verify our finding in Corollary 1.

## 5.3 Comparisons with Counterparts

Here, we conduct experiments on ImageNet-1K using ResNet-18 for comparing with existing post-normalization methods in terms of Top-1 accuracy and running speed, including B-CNN [31], LN [2], DeepO$_2$P [23], MPN-COV [29] with $\alpha = 0.5$, iSQRT-COV [28] and IB-CNN [30]. The results of all compared methods are listed in Table 6, where we can see that our DropCov outperforms plain GCP (i.e., GCP without normalization) by 2.4% (5.2%) for $d = 64$ ($d = 256$) with similar running time. Compared with element-wise post-normalization (i.e., LN, B-CNN and IB-CNN), our DropCov achieves significant performance gains while requiring less running time. For structure-wise post-normalization, our DropCov is remarkably superior to matrix logarithm normalization (DeepO$_2$P) in terms of both accuracy and running speed. Meanwhile, DropCov is superior or comparable to MPN-COV and iSQRT-COV using much less running time, especially during inference stage. Above results show our DropCov performs better or on par with the counterparts in terms of efficiency and effectiveness, providing an very promising normalization method for deep GCP networks.

## 5.4 Comparisons with Various Deep Architectures

**CNNs.** Here we apply our DropCov with $d = 128$ to various CNNs, including ResNet-34, ResNet-50 and ResNet-101. Meanwhile, we compare DropCov with the original CNNs on IN-1K, IN-C, IN-A and Sty.-IN. As shown in the upper part of Table 7, DropCov respectively brings about 2.6%, 2.2% and 1.8% gains over ResNet-34, ResNet-50 and ResNet-101 on IN-1K, while showing stronger robustness on IN-C, IN-A and Sty.-IN. Particularly, DropCov with ResNet-50 and ResNet-101 are respectively superior to the original ResNet-101 and ResNet-152, but having less model complexity.

**ViT.** For ViT models, we apply our DropCov with $d = 128$ to DeiT-S [43], Swin-T [32] and T2T-ViT-14 [50], and compare the original models as well as ConViT [6]. As shown in the bottom part of Table 7, DropCov brings about 2.6%, 1.3% and 1.2% gains over DeiT-S, Swin-T and T2T-ViT-14 on IN-1K, respectively. Meanwhile, it shows stronger robustness on IN-C, IN-A and Sty.-IN. Particularly, DropCov with DeiT-S is superior to DeiT-B and ConViT-B, but having less model complexity. Above results show our DropCov provides a simple yet effective method to improve deep architectures, and deep architectures with DropCov achieve better trade-off between accuracy and model complexity.

Table 7: Comparison of our DropCov with various ResNet and ViT models on four datasets, where results in terms of mCE and Top-1 accuracy (%) are reported on IN-C and reminding ones, receptively.

| Method | Params. | FLOPs | IN-1K (↑) | IN-C (↓) | IN-A (↑) | Sty.-IN (↑) |
|---|---|---|---|---|---|---|
| ResNet-34 [17] | 21.8 M | 3.66 G | 74.19 | 77.9 | 1.63 | 7.59 |
| ResNet-50 [17] | 25.6 M | 3.86 G | 76.02 | 76.7 | 2.47 | 7.15 |
| ResNet-101 [17] | 44.6 M | 7.57 G | 77.67 | 70.3 | 4.15 | 9.51 |
| ResNet-152 [17] | 60.2 M | 11.28 G | 78.13 | 69.3 | 5.98 | 10.09 |
| ResNet-34+DropCov (Ours) | 29.6 M | 5.56 G | $76.81_{(2.62)}$ | $71.1_{(6.8)}$ | $3.45_{(1.82)}$ | $11.16_{(3.57)}$ |
| ResNet-50+DropCov (Ours) | 32.0 M | 6.19 G | $78.19_{(2.17)}$ | $69.8_{(6.9)}$ | $5.08_{(2.61)}$ | $9.90_{(2.75)}$ |
| ResNet-101+DropCov (Ours) | 51.0 M | 9.90 G | $\mathbf{79.51}_{(1.84)}$ | $\mathbf{65.8}_{(4.5)}$ | $\mathbf{7.54}_{(3.39)}$ | $\mathbf{11.41}_{(1.90)}$ |
| DeiT-S [43] | 22.1 M | 4.6 G | 79.8 | 54.6 | 18.9 | 14.91 |
| Swin-T [32] | 28.3 M | 4.5 G | 81.2 | 62.0 | 21.6 | 13.40 |
| T2T-ViT-14 [50] | 21.5 M | 5.2 G | 81.5 | 53.2 | 23.9 | 15.80 |
| DeiT-B [43] | 86.6 M | 17.6 G | 82.0 | 48.5 | 27.4 | 17.94 |
| ConViT-B [6] | 86.5 M | 17.7 G | 82.4 | $\mathbf{46.9}$ | 29.0 | $\mathbf{19.67}$ |
| DeiT-S+DropCov (Ours) | 25.6 M | 5.5 G | $82.4_{(2.6)}$ | $52.6_{(2.0)}$ | $31.2_{(12.3)}$ | $17.10_{(2.19)}$ |
| Swin-T+DropCov (Ours) | 31.6 M | 6.0 G | $82.5_{(1.3)}$ | $54.8_{(7.2)}$ | $33.1_{(11.5)}$ | $14.13_{(0.73)}$ |
| T2T-ViT-14+DropCov (Ours) | 24.9 M | 5.4 G | $\mathbf{82.7}_{(1.2)}$ | $52.1_{(1.1)}$ | $\mathbf{31.7}_{(7.8)}$ | $18.81_{(3.01)}$ |

## 5.5 Comparisons on Long-tailed Benchmark

Finally, we transfer our DropCov models to iNat2017 [20], and compare with several deep architectures, i.e., ResNet-101, ResNet-152, Inception V3 (IncV3) [42] with SE [21] as well as the recently proposed ViT [10] and TransFG [16]. Specifically, we perform fine-tuning of our DropCov models with backbones of ResNet-101 and DeiT-S on iNat2017, namely DropCov (ResNet-101) and DropCov (DeiT-S). As shown in Table 8, our DropCov (ResNet-101) obtains 5.5% gains over the original ResNet-101, while outperforming ResNet-152 and IncV3 SE. Besides, our DropCov (DeiT-S) respectively outperforms ViT-B and TransFG by 3.7% and

Table 8: Comparisons (%) on iNat2017.

| Method | Top-1 | Top-5 |
|---|---|---|
| ResNet-101 | 62.4 | 84.1 |
| ResNet-152 | 64.2 | 85.5 |
| IncV3 SE | 66.3 | 86.7 |
| ViT-B_16 (IN-21K) [10] | 68.7 | N/A |
| TransFG (IN-21K) [16] | 71.7 | N/A |
| DropCov (ResNet-101) | 67.9 | 87.3 |
| DropCov (DeiT-S) | $\mathbf{72.4}$ | $\mathbf{90.3}$ |

0.7%, which are pre-trained on ImageNet-21K. The results clearly verify generalization of DropCov.

## 6 Conclusion

In this paper, we first analyze the effect of post-normalization on GCP from the perspective of optimizing deep GCP networks. Particularly, we find that effective post-normalization methods have a good ability to balance representation decorrelation and information preservation, which can reduce over-fitting and increase representation ability of deep GCP networks, respectively. According to our finding, we introduce some strategies to improve existing normalization methods, further verifying our finding. More importantly, we propose a novel pre-normalization for GCP (namely DropCov) based on adaptive channel dropout, which achieves very competitive performance with a linear complexity and training-only mode. Extensive experiments identify that our DropCov provides a simple yet effective solution to improve existing deep architectures on image classification tasks.

GCP normally produces a high-dimensional covariance representation, bringing a certain amount of computational cost, but our work makes use of high-dimensional GCP more flexible. Meanwhile, our DropCov can achieve clear gains for low-dimensional GCP with little extra computational cost (refer to Table 6). Additionally, our DropCov is potentially applicable to compact GCP methods [12, 24] and encourages exploration of other effective dropout strategies, which will be studied in future work.

## Acknowledgment

The work was sponsored by National Natural Science Foundation of China (Grant No.s 62276186, 61925602, 61971086 and 61732011), CCF-Baidu Open Fund (NO.2021PP15002000), CAAIXSJLJJ-2022-010C and Haihe Lab of ITAI (NO. 22HHXCJC00002).

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
