# Supplementary Materials for "DropCov: A Simple yet Effective Method for Improving Deep Architectures"

**Qilong Wang**[1,3]**, Mingze Gao**[1]**, Zhaolin Zhang**[1]**, Jiangtao Xie**[2]**, Peihua Li**[2]**, Qinghua Hu**[1,3,*]

[1]Tianjin University, China, [2]Dalian University of Technology, China,
[3] Haihe Laboratory of Information Technology Application Innovation, Tianjin, China
qlwang@tju.edu.cn, gaomingze@tju.edu.cn, zzl9@tju.edu.cn
jiangtaoxie@mail.dlut.edu.cn, peihuali@dlut.edu.cn, huqinghua@tju.edu.cn

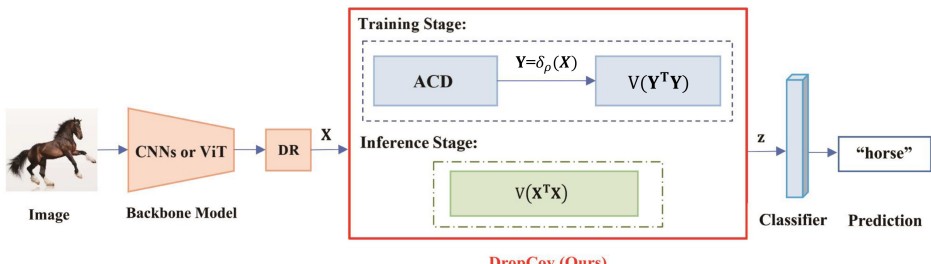

Figure S1: Overview of the proposed DropCov architecture for visual classification, where our DropCov can be flexibly integrated with existing deep convolutional neural network (CNNs) or ViT models. Please refer to Sec. S1 for the details.

## S1 Overview of DropCov Architecture

Our proposed DropCov can be flexibly integrated with existing deep architectures (e.g., CNNs [2] and ViT models [9, 8, 10]), whose flowchart is illustrated in Fig. S1. Specifically, an input image first passes through a backbone model (e.g., CNNs or ViT), and a dimensionality reduction (DR) module is inserted after the last convolution layer (or transformer block) of deep CNNs (or ViT). The DR module outputs $N$ $d$-dimensional features $\mathbf{X} \in \mathbb{R}^{N \times d}$, with which our DropCov replaces the original global average pooling or classification token to generate a covariance representation $\mathbf{z}$, which is fed into a classifier for final prediction. Particularly, during the training stage, our DropCov consists of an adaptive channel dropout (ACD) indicated by $\mathbf{Y} = \delta_\rho(\mathbf{X})$ and computation of covariance $(\mathbf{Y}^T\mathbf{Y})$ followed by vectorization and triangulation operations (V). For inference, we use $\mathrm{V}(\mathbf{X}^T\mathbf{X})$ without ACD for final prediction.

## S2 More Results for Effect of $\alpha$ on MPN

To further verify Corollary 1, we conduct more experiments to observe effect of power on matrix power normalization (MPN). Specifically, we experiment with ResNet-50 and ResNet-101 on ImageNet-1K and experiment with ResNet-18 on CIFAR100. Since MPN is very computationally expensive, we set $\alpha$ of MPN to $\{0.1, 0.3, 0.5, 0.7, 1.0\}$ and report the results (i.e., convergence curves) in Fig. S2 and

*Qinghua Hu is the corresponding author and is with Engineering Research Center of City intelligence and Digital Governance, Ministry of Education of the People's Republic of China.

36th Conference on Neural Information Processing Systems (NeurIPS 2022).

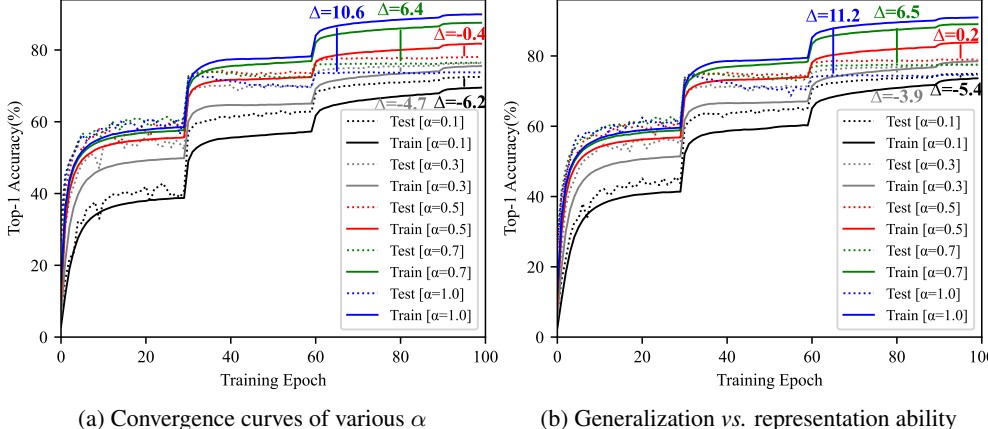

(a) Convergence curves of various $\alpha$

(b) Generalization *vs.* representation ability

Figure S2: Results of MPN-based GCP networks with various $\alpha$ using backbones of ResNet-50 and ResNet-101 ($d = 128$) on ImageNet-1K. (a) Convergence curves of GCP networks (ResNet-50) with various $\alpha \in (0, 1]$;(a) Convergence curves of GCP networks (ResNet-101) with various $\alpha \in (0, 1]$.

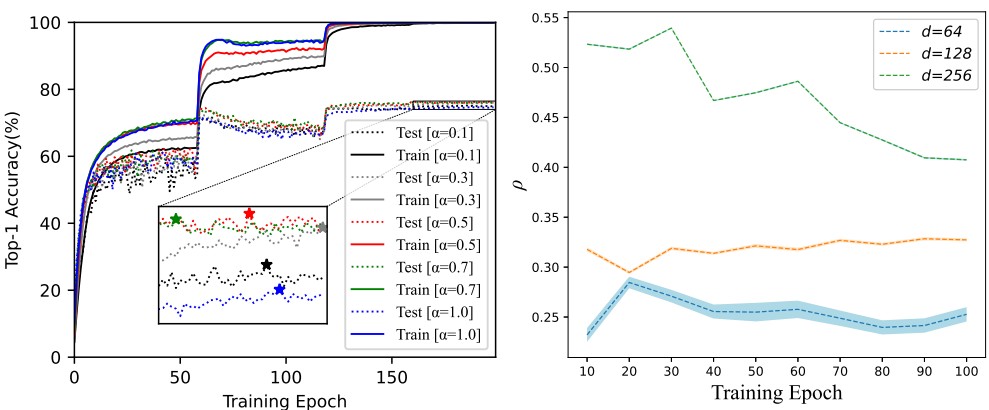

Figure S3: Results of GCP networks with matrix power normalization (MPN) using the backbone of ResNet-18 ($d = 128$) on CIFAR100.

Figure S4: Changes of $\rho$ for various $d$

Fig. S3, where $\Delta$ indicates the minimal gap between training and validation performance at a certain epoch of the last training stage. Particularly, we can see that behaviors of ResNet-50 and ResNet-101 with various $\alpha$ are consistent with those of ResNet-18 (i.e., Fig. 4 (a) of main manuscript), where $\alpha = 0.5$ achieves the best trade-off and so leads to the highest accuracies. For small-scale CIFAR100, all models achieve near perfect (100%) training accuracies after 150 epochs, and $\alpha = 0.5$ achieves the highest validation accuracy. Besides, [5, Fig. 3] (w.r.t. AlexNet on ImageNet) and [6, Fig.2] (w.r.t. VGG-VD on three small-scale fine-grained datasets) show 0.5 is the best choices of $\alpha$, and they have similar trends on recognition accuracies for various $\alpha$. These results suggest $\alpha$ of MPN has a consistent behavior for different models and various sizes of datasets, providing more supports on analysis on MPN in Corollary 1.

## S3  Analysis on Probability $\rho$ of ACD

Since our adaptive channel dropout (ACD) has the ability to adaptively determine probability $\rho$ of dropout, we illustrate behavior of probability $\rho$ for various feature dimensions along training epochs in Fig. S4, where ResNet-18 [2] is used as backbone and trained on ImageNet-1K [1]. Note that the dotted lines and shadow areas respectively represent mean and variance of $\rho$ for all training images, which demonstrate that our ACD can adaptively determine probability $\rho$ of dropout for various feature dimensions and inputs. Meanwhile, $\rho$ also varies along different training states (i.e.,

Table S1: Details of hyper-parameter settings of our DropCov models built with various deep architectures, which involve training-from-scratch on ImageNet-1K and fine-tuning on iNat2017.

| Architecture | Training-from-scratch on ImageNet-1K | | | | Fine-tuning on iNat2017 | |
|---|---|---|---|---|---|---|
| | ResNet | DeiT-S | T2T-ViT-14 | Swin-T | ResNet-101 | DeiT-S |
| Batch size | 256 | 896 | 896 | 1024 | 64 | 896 |
| Optimizer | SGD | AdamW | AdamW | AdamW | SGD | AdamW |
| Momentum | $\beta : 0.9$ | $\beta_1/\beta_2 :$ 0.9/0.95 | $\beta_1/\beta_2 :$ 0.9/0.95 | $\beta_1/\beta_2 :$ 0.9/0.95 | $\beta : 0.9$ | $\beta_1/\beta_2 :$ 0.9/0.95 |
| Epochs | 100 | 300 | 310 | 300 | 30 | 300 |
| Base learning rate | 1e-1 | 1e-3 | 5e-4 | 5e-4 | 4.5e-3 | 5e-5 |
| Final learning rate | 1e-4 | 1e-5 | 1e-5 | 1e-5 | 2.9e-3 | 1e-8 |
| Scheduler | step ($\times$0.1/30) | cosine | cosine | cosine | step ($\times$0.94/4) | cosine |
| Weight decay | 5e-4 | 0.03 | 0.03 | 0.05 | 0 | 0.03 |
| Label smoothing | - | 0.1 | 0.1 | 0.1 | - | 0.1 |
| Mixup | - | 0.8 | 0.8 | 0.8 | - | 0.8 |
| Cutmix | - | 1.0 | 1.0 | 1.0 | - | 1.0 |
| RandAugment | - | 9/0.5 | 9/0.5 | 9/0.5 | - | 9/0.5 |

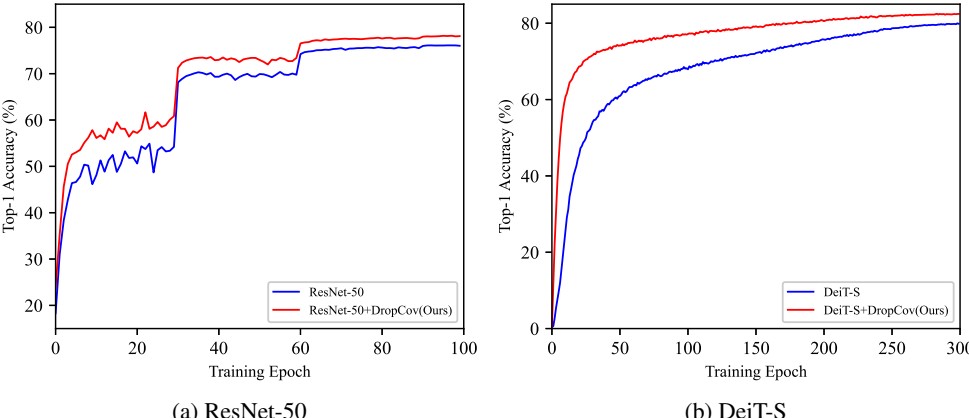

(a) ResNet-50      (b) DeiT-S

Figure S5: Convergence curve of our DropCov models with the backbones of ResNet-50 and DeiT-S on validation set of ImageNet-1K.

different training epochs). Besides, small feature dimension ($d = 64$) has larger variance of $\rho$ than large one ($d = 256$), which is caused by that feature correlation is closer related to its importance for small $d$, and so it is harder to select features for reaching the good trade-off. Meanwhile, larger feature dimension has smaller variance of $\rho$ for each epoch and ACD is more stable for all training samples. Above phenomena indicate our ACD can determine probability $\rho$ of dropout adaptively, and the results in Table 2 ∼ Table 5 of main manuscript show our ACD is clearly superior to those dropout methods with fixed $\rho$, verifying the effectiveness of our ACD.

## S4 Training Details

In this work, we apply the proposed DropCov to ResNet [2] and ViT models [9, 8, 10]. For training our DropCov models based on ResNet, we employ the same optimization policy and hyper-parameter settings as suggested in [5], whose details are listed in the first column of Table S1. For DropCov models based on ViT, we train them by following the configurations in [9, 8, 10], whose details are listed from the second column to the fourth one of Table S1. Particularly, Figs. S5a and S5b show convergence curve of our DropCov models with the backbones of ResNet-50 [2] and DeiT-S [9] on validation set of ImageNet-1K, respectively. From them we can see that DropCov models have faster convergence speed than the original deep CNNs and ViT models. Besides, we perform fine-tuning of

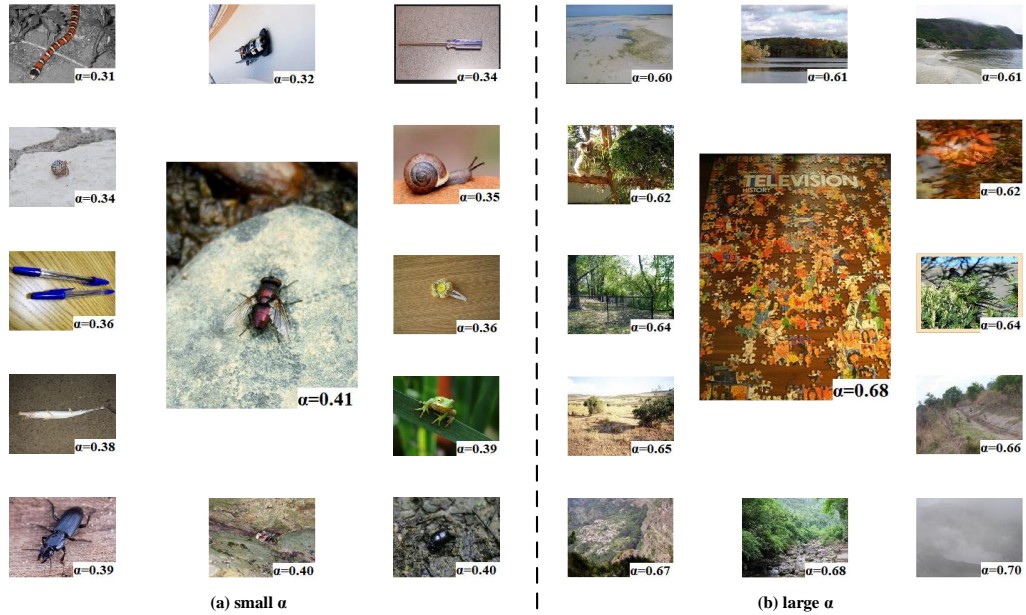

|     |     |
| --- | --- |
| (a) small α | (b) large α |

Figure S6: Visualization of some samples with large and small $\alpha$ achieved by APN on validation set of ImageNet-1K.

our DropCov models with backbones of ResNet-101 and DeiT-S on iNat2017 [3], whose details of hyper-parameter settings are given in the last two columns of Table S1.

## S5 More Evaluation of Modifications to Existing Methods

In this work, we make some modifications to existing post-normalization approaches according to our finding in Corollary 1. Particularly, we introduce an I-LogM (i.e., $\epsilon \log((\lambda_i + \epsilon)/\epsilon)$) to replace the original LogM (i.e., $\log(\lambda_i)$) for DeepO$_2$P [4]. Besides, a linear transform (LT) is introduced to B-CNN [7] and IB-CNN [6] for recovering information. Here, we assess effect of $\epsilon$ on I-LogM, while showing performance of plain GCP with only one LT module.

As listed in Table S2, we can see that single LT module brings a little gain for plain GCP. Compared to B-CNN + LT (79.62% training accuracy), plain GCP + LT (91.26% training accuracy) suffers much heavier over-fitting for large feature dimension (i.e., $d = 256$) due to no representation decorrelation involved in plain

Table S2: Results (Top-1 accuracy in %) of some modifications to existing post-normalization methods using ResNet-18 on ImageNet-1K.

| Method | $d = 64$ | $d = 256$ |
| --- | --- | --- |
| Plain GCP | 71.1 | 70.0 |
| Plain GCP + LT | 71.2 | 70.6 |
| B-CNN [7] | 38.3 | 41.1 |
| B-CNN + LT | 68.3 | 73.2 |
| DeepO$_2$P [4] | 70.1 | *Not Converge* |
| LogM (+ 1e-3·**I**) | 70.6 | 67.0 |
| I-LogM ($\epsilon$=1e-3) | 70.8 | 71.1 |
| I-LogM ($\epsilon$=1) | 71.2 | 72.0 |
| DropCov (Ours) | **73.5** | **75.2** |

GCP + LT, while B-CNN + LT achieves significant improvement over B-CNN and plain GCP. These results indicate that good trade-off between representation decorrelation and information preservation plays a key role in effectiveness of post-normalization for GCP. For modifying LogM, a widely used solution is adding a small value to diagonal elements of covariance matrix (e.g., $\mathbf{X}^T\mathbf{X}$+ 1e-3·**I**) [5, 6], which is indicated by LogM (+ 1e-3·**I**). Different from them, we directly handle eigenvalues using a shift operation to make all modified eigenvalues (i.e., inputs of $\log$) be larger than one. As compared in Table S2, I-LogM ($\epsilon$=1e-3) is superior to LogM (+ 1e-3·**I**), especially for large feature dimension (i.e., $d = 256$). Meanwhile, $\epsilon$=1 brings further performance gains, which is used in our experiments.

## S6 Visualization of $\alpha$ of APN

To further analyze our introduced adaptive power normalization (APN), we pick up some samples with large and small $\alpha$ achieved by APN on validation set of ImageNet-1K, which are illustrated in Figs. S6. From them we can observe that the samples containing simple objects and more redundant information have small $\alpha$, where MPN tends to representation decorrelation. On the contrary, the samples involving less redundant information (e.g., scene) have large $\alpha$, where MPN tends to information preservation. Such these phenomena show the consistency with our finding.