# OpenReview forum: "DropCov: A Simple yet Effective Method for Improving Deep Architectures"
_NeurIPS.cc/2022/Conference — NeurIPS 2022 Accept_

### Official Review · Reviewer_VWPn · 2022-07-10

**Rating:** 4
**Confidence:** 4
**Soundness:** 2 fair
**Presentation:** 3 good
**Contribution:** 3 good

**Summary:**

This paper focuses on studying matrix power normalization in global covariance pooling operation. The authors provide a detailed analysis in the effect of different power parameters on matrix power normalization as well as other post normalizations, in terms of feature decorrelation and information preservation, empirically showing the necessary of using a good tradeoff between the two properties. To reduce the computation complexity of post normalization, the authors introduce Dropconv which performs channel dropout to achieve feature decorrelation. Experiments show the ability of the proposed strategy  on performance improvements on both ResNet and ViT backbones.

**Questions:**

1. For the experiments in Table 2, the counterpart of DropChannel seems analogous to ACD except that the probability in ACD is determined by features, while the performance gap is so significant and even higher than DropElement. Could the authors analyze the phenomenon? How does the DropChannel and DropElement perform when using different dropout probability beyond 0.5?

2. It is unclear to me why computational complexity is linear for DropCov while quadratic to element-wise methods (in Table 1).


**Limitations:**

The paper analyzes post-normalization for global covariance pooling unit and introduces channel dropout to improve it, which is not restricted to certain downstream cv tasks. It seems like no potential negative societal impact.

**Strengths And Weaknesses:**

Strengths:
1. The authors provide a detailed analysis in the effect of the power parameter of matrix power normalization on feature decorrelation and information preservation, and consequential impact on model performance empirically, which is valuable for related research area.

2. The paper is generally well written and easy to follow, although some statements need to be explained and clarified.

3. Extensive experiments on representative traditional and modern backbones are conducted to validate the method.

Weaknesses:
1. Some descriptions are less rigorous and need to be clarified:

The first corollary (line 120) states that setting power parameter 0.5 is optimal according to Eq. 2 and Fig. 1, while it lacks either theoretical proof or comprehensive empirical studies. Is the behavior of power parameter fairly invariant to dataset or network backbone?

For the proposed Adaptive Power Normalization, the power parameter is optimized according to eigenvalues of feature covariance, while the features are updated during training. How does it realize in practice? There is a similar question for the dropout probability in Eq. 5. Could the authors provide their training behavior if they are variable during training? If the parameters are adaptive with model training from scratch, primitive parameters may be far from optimal values. On the other hand, if the variance of probability distribution is rather small, it may indicate less necessary for tuning it during training.

2. The motivation of performing dropout on convolution channels is not new. The main difference is specializing the strategy with global covariance pooling.

3. Equipping DropCov into baseline backbones (e.g., ResNet50) brings significant increase on parameter size and computational overhead.

---

> ### Author Response · Authors · 2022-08-02
> **Response to the comments of Reviewer VWPn （Part I）**
>
> We sincerely thank the reviewer for recognizing the detailed and valuable analysis of MPN, and positive comments on good writing and extensive experiments. In the following, we respond carefully to the reviewer’s questions and hope our answers could address your concerns.
>
> [W1-1]: The first corollary (line 120) states that setting power parameter 0.5 is optimal according to Eq. 2 and Fig. 1, while it lacks either theoretical proof or comprehensive empirical studies. Is the behavior of power parameter fairly invariant to dataset or network backbone?
>
> [A]: Thanks for the comments. Kindly note that theoretical proof on optimal power parameter is still an open problem, while [27] shows power parameter 0.5 accounts for a robust covariance estimation. For empirical studies, both Fig. 3 (w.r.t. AlexNet on ImageNet) in [27] and Fig. 2 (w.r.t. VGG-VD on three small-scale datasets) in [28] show 0.5 is the best choices of $\alpha$, and they have similar trends on accuracies for various $\alpha$. Additionally, we conduct more experiments (please refer to [Common Q2] above) to observe effect of $\alpha$ of MPN, which show $\alpha$ of MPN has a consistent behavior for different backbones. Above results suggest behavior of power parameter is invariant to dataset or network backbone.
>
> [W1-2]: How do adaptive power normalization (APN) and Eq. (5) realize in practice? Could the authors provide their training behavior if they are variable during training?
>
> [A]:  Thanks for the comments. For realizing APN, we can use the same computational process with MPN [27], except choice of $\alpha$. Given an input $\mathbf{X}$, we first compute covariance $\mathbf{X}^{T}\mathbf{X}$ and obtain its eigenvalues {$\lambda_{i}$} using eigenvalues decomposition. Based on {$\lambda_{i}$}, our APN computes $\alpha$ by solving Eq. (4) with a grid search strategy. Finally, we perform regular MPN using the optimized $\alpha$. To realize APN in Eq. (5), we first compute $\boldsymbol{\omega}$ and $\boldsymbol{\pi}$ with input $\mathbf{X}$ using the bottom part of Eq. (5). Then, $\rho$ can be easily calculated by using the upper part of Eq. (5) with $\boldsymbol{\omega}$ and $\boldsymbol{\pi}$. Finally, we perform channel dropout on input $\mathbf{X}$ using $\rho$, and then compute GCP. All source code will be released once acceptance. For training behavior, please refer to [Common Q1] above for the details.
>
> [W2]: The motivation of performing dropout on convolution channels is not new. The main difference is specializing the strategy with global covariance pooling.
>
> [A]: Kindly note our DropCov is motivated by the finding in Corollary 1, and try to achieve good trade-off between representation decorrelation and information preservation more efficiently. To our best knowledge, we make the first attempt to develop dropout-based method (i.e., ACD) for normalizing GCP. Besides, our ACD is clearly different from existing dropout methods (e.g., maxdropout and DropConnect) those require manual tuning of probability $\rho$, ACD has the ability to adaptively determine probability $\rho$ of dropout for reaching a good trade-off between representation decorrelation and information preservation, while being clearly superior to channel dropout with fixed dropout ratios (see Tab.2 and [Common Q3] above).
>
> [W3]: DropCov brings significant increase on parameter size and computational overhead.
>
> [A]: We would like to clarify that DropCov actually achieves better trade-off between performance and model complexity, while bringing affordable computational complexity especially inference speed. As shown in Tab. 5, ResNet-50 and ResNet-101 with DropCov respectively achieve better performance than ResNet-101 and ResNet-152 with GAP, while having much less model complexity (e.g., 32.0M parameters and 6.19G FLOPs of ResNet-50+DropCov vs. 44.6M parameters and 7.57G FLOPs of ResNet-101). Besides, as shown in Tab. 6, ResNet-101 with GCP is much lightweight and better than ResNet-152 with GAP on iNat2017.  Additionally, the results in Tab. 4 show our DropCov with $d=64$ (73.5\%) achieves clear gains over GAP (70.2\%) using nearly inference time (1.04$\mu$s vs 0.97$\mu$s), where only extra <1.0M parameters and 1.0 GFLOPs are introduced. It indicates DropCov with low-dimensional GCP (e.g., $d=64$) still brings clear improvement over GAP, while introducing moderate computational cost. Finally, since our DropCov performs no normalization during inference, it achieves comparable inference speed with GAP (refer to table in https://github.com/36f857fe/InferSpeed), especially for strong ResNet-101 and ViT models.

---

> > ### Author Response · Authors · 2022-08-02
> > **Response to the comments of Reviewer VWPn （Part II）**
> >
> > [Q1]: For the experiments in Table 2, the counterpart of DropChannel seems analogous to ACD except that the probability in ACD is determined by features, while the performance gap is so significant and even higher than DropElement. Could the authors analyze the phenomenon? How does the DropChannel and DropElement perform when using different dropout probability beyond 0.5?
> >
> > [A]: Thanks for the comment. We experiment with DropChannel and DropElement by using different dropout ratios (please refer to [Common Q3] above). <1> In Tab. 2, DropChannel and DropElement are performed with the fixed dropout probability of 0.5. Note that for the same dropout probability, DropChannel drops about two times of elements than DropElement for covariance representations. Therefore, for small feature dimensions (i.e., $d=64$), dropout probability of 0.5 is too large for DropChannel, leading an inferior performance. As shown in [Common Q3], DropChannel (dim=64) with $\rho=0.3$ achieves a comparable result. Besides, it is harder to select channel features for reaching the good trade-off for small feature dimension, where feature correlation is closer related to its importance. Therefore, for dim=64, DropChannel with $\rho=0.3$ is slightly inferior to DropElement with $\rho=0.5$. For large feature dimension (i.e., $d=256$), DropChannel with the best $\rho$ of 0.5 is superior to the best DropElement ($\rho=0.7$). <2> As shown in [Common Q3], ACD is superior to DropElement and DropChannel for all dropout ratios, since our ACD can adaptively determine probability $\rho$ of dropout by balancing representation decorrelation and information preservation. These results further demonstrate the effectiveness of our ACD.
> >
> >
> > [Q2]: It is unclear to me why computational complexity is linear for DropCov while quadratic to element-wise methods (in Table 1).
> >
> > [A]: Thanks for the comment. Table 1 compares computational complexity of different normalization approaches. The normalization of DropCov is to perform dropout on channel of features (i.e., which channel is dropped or not), whose complexity is linear with respect to number of channels (i.e., $d$). For element-wise normalization methods, they perform normalization on each element of covariance representations. For $d$-dimensional features $\mathbf{X}$, dimension of covariance representations is $d^{2}$, and so the computational complexity of element-wise normalization methods is $O(d^{2})$.

---

### Official Review · Reviewer_5eq4 · 2022-07-11

**Rating:** 5
**Confidence:** 3
**Soundness:** 3 good
**Presentation:** 3 good
**Contribution:** 2 fair

**Summary:**

This paper focuses on the post-normalization of the pooling strategy leveraging second-order statistics, i.e., global covariance pooling (GCP).  The key contributions are twofold: 1) the paper shows that post-normalization in matrix power normalization [27] tries to balance representation decorrelation and information preservation through empirical analysis; 2) the paper proposes a linear complexity pre-normalization DropCov of GCP based on an adaptive channel dropout.

The proposed approach has demonstrated its effectiveness on several image classification benchmarks including ImageNet, ImageNet-C, ImageNet-A, stylized ImageNet, and long-tailed classification with different backbones (CNN, transformer).


**Questions:**

Please refer to the weaknesses.


**Limitations:**

No potential negative societal impact.

**Strengths And Weaknesses:**

**Strength**:
* [S1]: I think the main strength of this work is the experiments. The paper provides diverse experiments on different datasets with different backbones to validate the proposed DropCov.

* [S2]: The paper is presented very well and reads very well. I read quite carefully but did not find any typos. The paragraphs and sections are also self-contained and organized well.

* [S3]: The analysis of MPN (Sec.2.1) is intuitive, and the empirical results are interesting.

**Weakness**:
* [W1]: I think the main drawback of this work is the significance. I can acknowledge the improvement is clear, the idea is simple. But as GCP is computationally more expensive than 1st order pooling, not much attention is paid to the field.

* [W2]: I think the analysis on MPN is a little shallow. the experiments are on a single dataset with a network. It would be interesting to see the effect of power on different sizes of datasets with different capacities of models.

* [W3]: For the adaptive power normalization, the improvement with the proposed formulation is quite marginal. Moreover, I am wondering whether extra visualization would provide some insight. For example, it might be possible to visualize samples with important/small alpha to verify corollary 1 further.

* [W4]: Equation 5 is complicated and not intuitive. For example, the cosine similarity between the feature importance and feature correlation is weird. Also, this similarity might be negative and cause the probability larger than 1? Equation 6 might be confused as well. The dropout should be only applied once on $X$ (not separately for $X$ and $X^T$)?

* [W5]: It would be more convincing to conduct different dropout ratios (rather than a fixed ratio of 0.5) to validate the effect of adaptive dropout.

* [W6]: It is not apparent to understand the intuition behind the experiments of “element” and “channel” dropout. For example, in lines 275-276, the paper emphasizes applying one type of dropouts is equivalent to considering only feature correlation or feature importance. It would be better to provide a more detailed explanation on this part.

---

> ### Author Response · Authors · 2022-08-02
> **Response to the comments of Reviewer 5eq4**
>
> We sincerely thank the reviewer for positive comments on strength of experiments, good presentation, and interesting results about analysis of MPN. In the following, we respond carefully to the reviewer’s questions and hope our answers could address your concerns.
>
> [W1]: I think the main drawback of this work is the significance. I can acknowledge the improvement is clear, the idea is simple. But as GCP is computationally more expensive than 1st order pooling, not much attention is paid to the field.
>
> [A]: Thanks for the comment. We kindly note deep GCP has been successfully adopted to various tasks (e.g., fine-grained/general image classification, video recognition, ReID, few-shot learning and Graph NN), and a mount of works have been published on top-tier conferences and journals (a brief summary could be found in a public website: https://saimunur.github.io/spd-archive/ under the umbrella of "symmetric positive definite matrices"). It indicates GCP has attracted a lot of research interests. Meanwhile, we would like to clarify GCP actually achieves better trade-off between performance and computational complexity than 1st order pooling (GAP). Specifically, as shown in Tab. 5, ResNet-50 and ResNet-101 with DropCov respectively achieve better performance than ResNet-101 and ResNet-152 with GAP, while having much less computational complexity. The similar phenomena also are observed for ViT models. Besides, as shown in Tab. 6, ResNet-101 with GCP is much lightweight and better than ResNet-152 with GAP on long-tailed iNat2017. Additionally, since our DropCov performs no normalization during inference, it achieves comparable inference speed with GAP (refer to table in https://github.com/36f857fe/InferSpeed), especially for strong ResNet-101 and ViT models.  In conclusion, previous works have shown that GCP is a very competitive option in a variety of visual tasks, as compared to GAP, and our DropCov provides a more efficient yet effective solution.
>
>
> [W2]: Effect of power on different sizes of datasets with different capacities of models.
>
> [A]: Thanks for the comments. Please refer to [Common Q2] above for more results.
>
>
> [W3]: Improvement of APN is quite marginal & extra visualization
>
> [A]: Kindly note our APN is proposed for verifying the claim in Corollary 1. As stated in line 284-285, average values of $\alpha$ achieved by APN are nearby 0.5, which further account for why 0.5 is the widely used choice of $\alpha$ for MPN. Besides, APN brings 0.1%-0.2% gains over MPN with $\alpha=0.5$ by considering effect of inputs. We would like to clarify they are non-trivial gains on ImageNet over strong MPN, and the recent works [r1,r2] also bring similar gains (0.1%-0.3%) over MPN. These results verify the conclusion in Corollary 1. Furthermore, we pick up some samples with large and small $\alpha$ achieved by APN on validation set of ImageNet-1K, which are shown in an anonymous URL: https://github.com/36f857fe/Visualization. From them we can observe the samples containing simple objects and more redundant information have small $\alpha$, where MPN tends to representation decorrelation. On the contrary, the samples involving less redundant information (e.g., scene) have large $\alpha$, where MPN tends to information preservation. Such these phenomena show the consistency with our finding.
>
> [r1] Why Approximate Matrix Square Root Outperforms Accurate SVD in Global Covariance Pooling? ICCV, 2021
>
> [r2] Improving Covariance Conditioning of the SVD Meta-layer by Orthogonality. ECCV, 2022
>
> [W4]: Discussion on Eqn (5) & range of $\rho$ in Eqn (5) & dropout in Eqn (6)
>
> [A]: Thanks for the comments . <1> Please refer to [Common Q1] above for more discussions on Eqn (5). <2> In practice, we restrict $\rho$ in Eqn (5) by using $\max(0, \rho)$ and $\min(1, \rho)$. <3> Thanks for pointing out this potential ambiguity, and we will modify Eqn (6) to $\mathbf{z}=\textrm{V}(\mathbf{Y}^{T}\mathbf{Y}), \mathbf{Y}=\delta_{\rho}(\mathbf{X})$.
>
>
> [W5]: Experiments with different dropout ratios
>
> [A]: Thanks for the comments. Please refer to [Common Q3] above for more results.
>
>
> [W6]: The intuition behind “element” and “channel” dropout.
>
> [A]: Thanks for the comment. The “element” dropout preforms dropout on elements of covariance representations based on values of elements ($[\mathbf{X}^{T}\mathbf{X}]_{ij}$), which indicates correlation between $i$-channel feature and $j$-channel feature. Therefore, “element” dropout only considers feature correlation. For “channel” dropout, it preforms dropout on feature channels based on channel weights, which is obtained by attention module of ACD (i.e., $\boldsymbol{\omega}$). Therefore, “channel” dropout only considers feature importance. Based on above discussion, “element” dropout and “channel”dropout can be regarded as one type of dropouts is equivalent to considering only feature correlation or feature importance, respectively. We will add above explanations in the revision.

---

### Official Review · Reviewer_h5SV · 2022-07-18

**Rating:** 7
**Confidence:** 3
**Soundness:** 3 good
**Presentation:** 2 fair
**Contribution:** 3 good

**Summary:**

Motivated by the facts of
(1) the post-normalization of GCP lacks theoretical understanding and
(2) the high computational complexity of GCP normalization,
the paper proposes:
- a theoretical analysis of matrix power normalization in the GCP networks and extend it to an adaptive normalization method
- an adaptive channel dropout to reduce the computational complexity.

The proposed method has been verified on ImageNet and iNaturalist datasets using various scale architectures.

**Questions:**

- Is the GCP the bilinear pooling? I think it could be better to explain these two academic terms and illustrate their relationships in the paper.

**Limitations:**

None.

**Strengths And Weaknesses:**

Strengths
- The proposed method is simple and efficient for classification tasks.
- Experiments are solid on multiple datasets, especially have experimented with different scale architectures.
- Good theory analysis with ablation studies.

---

> ### Author Response · Authors · 2022-08-02
> **Response to the comments of Reviewer h5SV**
>
> We sincerely appreciate the reviewer for positive reviews on efficient method, soild experiments, good theory analysis and support to our paper. In the following, we respond carefully to the reviewer’s question and hope our answer could address your concerns.
>
> [Q1]: Is the GCP the bilinear pooling? I think it could be better to explain these two academic terms and illustrate their relationships in the paper.
>
> [A]: Thanks for the comment. GCP and bilinear pooling share very similar mathematical formulas, but there exist some differences between them. Specifically, GCP focuses on computing covariance of inputs $\mathbf{X}$ with mean of $\boldsymbol{\mu}$ (i.e., $(\mathbf{X}-\boldsymbol{\mu})^{T}(\mathbf{X}-\boldsymbol{\mu})$), while bilinear pooling computes the outer product of inputs $\mathbf{X}$ and $\mathbf{Y}$ (i.e., $\mathbf{X}^{T}\mathbf{Y}$). When $\mathbf{X}$ and $\mathbf{Y}$ are shared (a most widely used case for bilinear pooling) and the inputs are zero-mean, bilinear pooling captures the same information with one of GCP. Besides, we note that element-wise signed sqrt-root followed by a $\ell_{2}$ normalization is the default post-normalization for bilinear pooling ($\mathbf{X}^{T}\mathbf{X}$)  in the original paper [29]. In this work, we focus on effect of different post-normalization approaches for GCP, and compared them in Table 4. In the revision, we will add above discussions to illustrate relationships between GCP and bilinear pooling.

---

### Official Review · Reviewer_Fzo6 · 2022-07-18

**Rating:** 5
**Confidence:** 3
**Soundness:** 3 good
**Presentation:** 2 fair
**Contribution:** 2 fair

**Summary:**

The main contribution of this paper is to propose Adaptive Channel Dropout (ACD), which performs channel dropout before the covariance computation to obtain good trade-off between representation decorrelation and information preservation. ACD is more efficient and more effective than existing  post-normalization techniques. Since the core of ACD is dropout, it is only used during training, and is not needed in inference. There are other minor contributions, such as Adaptive Power Normalization (APN), which is a slight improvement over matrix power normalization (MPN) [34,42]. The paper also performs more analysis of MPN.

**Questions:**

 - more (theoretical) discussion of equation (5)
 - clarify contribution and even rewrite the paper to make the core contribution more obvious
 - comparison to other training regularization techniques that prevent overfitting

**Limitations:**

Checklist points to the conclusion, but the conclusion does not really address the limitations and potential negative societal impact of the work.

**Strengths And Weaknesses:**

Strengths
 - ACD is more effective and efficient than existing post-normalization techniques.
 - more extensive analysis of alpha for MPN

Weaknesses
 - One of the main contribution is equation (5), but the intuition/explanation is not very clear. Some more theoretical discussion would also be helpful.
 - The current presentation makes it hard to understand what are the main contributions. There are many things going on: analysis of MPN, LogM. APN. ACD... My take is ACD is the main contribution, but would be helpful to make it clear in the paper
 - The main advantages of ACD seems to come from the fact that it is a dropout-based technique. The fact that its faster and is not required in inference are because it is based on dropout, which is quite different from post-norm/pre-norm. It would be helpful to compare to more regularization techniques in training.

---

> ### Author Response · Authors · 2022-08-02
> **Response to the comments of Reviewer Fzo6**
>
> We sincerely thank the reviewer for recognizing efficiency and effectiveness of our ACD as well as extensive analysis of MPN. In the following, we respond carefully to the reviewer’s questions and hope our answers could address your concerns.
>
> [W1&Q1]: more (theoretical) discussion of equation (5)
>
> [A]:  Thanks for the comments. Please refer to [Common Q1] above.
>
>
> [W2&Q2]: clarify contribution and even rewrite the paper to make the core contribution more obvious
>
> [A]: Thanks for the comments. We would like to clarify that our core contributions consist of two parts: (1) we make the first attempt to understand effect of post-normalization on deep GCP from the perspective of model training (i.e., Sec. 2) and (2) we propose an efficient and effective DropCov method for normalizing GCP (i.e., Sec. 3). Meanwhile, these two contributions are closely related. Specifically, for the first contribution, Sec. 2.1 takes matrix power normalization (MPN) as an example, and concludes that effective post-normalization can make a good trade-off between representation decorrelation and information preservation for GCP, which are crucial to alleviate over-fitting and increase representation ability of deep GCP networks, respectively. Then, Sec. 2.2 aims to further verify the finding in Sec. 2.1 by introducing APN and extending the finding on MPN to other post-normalization approaches (e.g., LogM and EwN). Therefore, APN and analysis of LogM in Sec. 2.2 provide further support on the conclusion in Sec. 2.1, while combination of Sec. 2.1 and Sec. 2.2 provides a full view for effect of existing post-normalization methods. For the second contribution, our DropCov develops an efficient and effective ACD for normalizing deep GCP, which is strongly motivated by the finding in Sec. 2 (the first contribution), i.e., how to achieve good trade-off between representation decorrelation and information preservation more efficiently. In summary, Sec. 2 and Sec. 3 respectively describe our two core contributions, where the finding in Sec. 2 encourages us to propose the method in Sec. 3.
>
> As suggested by the reviewer, we will further compress the space of Sec.2.2 and give more discussions on our ACD in Sec. 3 to highlight contribution of our DropCov in the revision.
>
> [W3&Q3]: comparison to other training regularization techniques that prevent overfitting
>
> [A]: Thanks for the suggestion. The regularization perspective for post-normalization of GCP is interesting, and both existing post-normalization approaches and our ACD can regarded as performing some regularization strategies on features or covariances. Particularly, GCP with structure-wise post-normalization can be rewritten as $\mathbf{Z}=f(\mathbf{X})f(\mathbf{X})^{T}, s.t. \min_{f(\mathbf{X})}\|f(\mathbf{X})f(\mathbf{X})^{T}-P(\mathbf{X}^{T}\mathbf{X})\|$, where $P(\mathbf{X}^{T}\mathbf{X})$ are $(\mathbf{X}^{T}\mathbf{X})^{\alpha}$ and $\log(\mathbf{X}^{T}\mathbf{X})$ for MPN and LogM, respectively. For our ACD, $f(\mathbf{X})=\delta_{\rho}(\mathbf{X})$. But note that our work for the first time shows how do existing post-normalization approaches work for optimizing GCP networks. As suggested by the reviewer, we compare with several regularization techniques (i.e., Maxout [r1], DropConnect [r2], Decov [r3] and maxdropout [r4]) following the settings in Tab. 4. As shown in below table, our ACD clearly outperforms other methods. Although these methods can prevent overfitting, they are not good at balancing representation decorrelation and information preservation. Above results further verify the effectiveness of our ACD. In the revision, we will compare with more regularization techniques, including more variants of dropout (e.g., DropBlock [NeurIPS18]) and other weight regularization strategies.
>
> | Method | Maxout  | Dropconnect |  Decov  |  Maxdropout  | ACD (Ours)  |
> | :----: | :----: | :----: | :-----:| :----: | :----: |
> | Top-1 Acc(%) d=64 |   72.11   |    70.59   | 72.42      |     71.95    |    73.50 |
> | Top-5 Acc(%) d=64 |   90.56   |    89.34   | 90.69       |     90.11    |    91.36  |
> | Top-1 Acc(%) d=256|   73.67   |    72.46   | 74.01    |     70.11    |    75.20  |
> | Top-5 Acc(%) d=256|   91.37   |    90.23   | 91.55    |     88.97    |    92.13  |
>
> [r1] Maxout networks. In ICLR, 2013.
>
> [r2] Regularization of neural networks using dropconnect. In ICML, 2013.
>
> [r3] Reducing Overfitting in Deep Networks by Decorrelating Representations. In ICLR, 2016.
>
> [r4] Maxdropout: Deep neural network regularization based on maximum output values. In ICPR, 2020.

---

### Author Response · Authors · 2022-08-02
**Response to some common questions**

We sincerely thank the reviewers for the valuable comments. Here, we response to three common questions and hope our answers could address your concerns.

[Common Q1]: More discussion of equation (5)

[A]: For realizing our ACD, Eqn (5) aims to adaptively decide probability $\rho$ of channel dropout for reaching a good trade-off between representation decorrelation and information preservation, where $\frac{D}{\log(d)}$ and inner product of <$\boldsymbol{\omega},\boldsymbol{\pi}$> are designed to consider effect of feature dimension ($d$) and relationship between feature correlation and feature importance, respectively. In particular, feature correlation (i.e., $\boldsymbol{\pi}$ computed by summarizing the elements along row of $\mathbf{X}^{T}\mathbf{X}$) and feature importance (i.e., $\boldsymbol{\omega}$ achieved by channel attention) indicate representation decorrelation and information preservation, respectively. Clearly, larger feature correlation results in stronger representation correlation, while features with larger channel weights contain more important information. Therefore, relationship between feature correlation and feature importance is good indicator for performing channel dropout. Intuitively, if feature correlation is close related to (i.e., closely decoupled with) feature importance, it is hard to select features for reaching a good trade-off between representation decorrelation and information preservation, and so we need to carefully adopt dropout for channel features under the random setting, leading to a small $\rho$. Otherwise, we can perform dropout more safely to achieve a trade-off, and adopt a large $\rho$.

Besides, we have showed behavior of $\rho$ during training in Fig. A2 of supplementary materials, where we observe $\rho$ varies along training epochs, and $\rho$ has a clear variance for all training samples at each epoch. They show ACD has the ability to adaptively determine probability $\rho$ of dropout. Besides, small feature dimension $(d=64)$ has larger variance of $\rho$ than large one $(d=256)$, which is caused by feature correlation is closer related to its importance for small $d$, and so it is harder to select features for reaching the good trade-off. The results in Tabs. 2 and 3 show our ACD is clearly superior to those with fixed $\rho$, verifying the effectiveness of our ACD.

We will add more discussion of Eqn (5) in the revision.

[Common Q2]: Effect of power on different sizes of datasets with different capacities of models

[A]: We further experiment with ResNet-50 and ResNet-101 on ImageNet-1K to observe effect of power. Since MPN is very computationally expensive (see Tab. 4), we set $\alpha$ of MPN to {0.1, 0.3, 0.5, 0.7, 1.0} and report the results (i.e., convergence curves) achieved before deadline in an anonymous URL: https://github.com/36f857fe/PowerMPN, where we can see that behaviors of ResNet-50 and ResNet-101 with various $\alpha$ are consistent with those of ResNet-18 (i.e., Fig. 1 (a)), verifying our analysis on MPN again. Besides, we kindly note that both Fig. 3 (w.r.t. AlexNet on ImageNet) in [27] and Fig. 2 (w.r.t. VGG-VD on three small-scale datasets) in [28] show 0.5 is the best choices of $\alpha$, and they have similar trends on recognition accuracies for various $\alpha$. These results suggest $\alpha$ of MPN has a consistent behavior for different models and various sizes of datasets.

We will report more results on effect of $\alpha$ in the revision.

[Common Q3]: 'DropElement' and 'DropChannel' with different dropout ratios

[A]: We experiment with 'DropElement' and 'DropChannel' by using different dropout ratios, where we adopt exactly the same settings in Tab. 2. Due to time limitation, we set feature dimension $d$ to 64 and 256, while dropout ratios vary in the range of $[0.1, 0.3, 0.5, 0.7, 0.9]$. As compared in below table, our ACD is superior to 'DropElement' and 'DropChannel' for all dropout ratios. Particularly, the best dropout ratios are quite different for two methods (i.e., 'DropElement' and 'DropChannel') and various feature dimensions. In contrast, our ACD can adaptively determine probability $\rho$ of dropout by balancing representation decorrelation and information preservation, while always achieving the best performance. These results further demonstrate the effectiveness of our ACD.

|   Method   | | |Channel |Dropout   | |  Element |Dropout   |
| :----: | :----: | :----: |:----: | :----: | :----: | :----: | :----: |

|     $ρ$    |  dim = 64 | dim = 256 |  dim = 64 | dim = 256 |
| :----: | :----: | :----: |:----: | :----: |
|     0. 1    |   72.5   |   71.9   |   72.3   |   70.8   |
|     0. 3    |   72.8   |   74.7   |   72.8   |   72.3   |
|     0. 5    |   70.1   |   75.1   |   73.4   |   74.0   |
|     0. 7    |   65.7   |   72.1   |   72.1   |   74.2   |
|     0. 9    |   20.5   |   54.7   |   68.2   |   73.8   |
|  ACD (Ours) |   73.5   |   75.2   |   73.5   |   75.2   |

---

### Meta-Review · Area_Chair_NuB2 · 2022-08-30

**Recommendation:** Accept
**Confidence:** Less certain

**Metareview:**

Both reviewer Fzo6, reviewer VWPn and reviewer 5eq4 have concerns and been questions regarding equation 5. Please clarify the clarifications on the paper and add intuition and more discussion of Eq. 5.

The paper and comments from the authors indicate that dropout base regularizations are effective (Maxdropout, Maxout and Decov also outperforms GCP). This does mean that a large part of the benefits of the proposed method (no inference processing, lower complexity) are the result of that dropout/regularization in training, which takes away from the contributions.

Overall, I think the paper is borderline, leaning to acceptance, as the proposed DropConv does out perform other dropout/regularization methods and I believe the paper might benefit the community. I'd strongly encourage the authors to review their manuscript and address the reviewers’ concerns as best as possible in the revised manuscript.

**Award:**

No

---

### Decision · Program_Chairs · 2022-09-14

Accept